# ON THE STABILITY OF MULTI-BRANCH NETWORK

## ABSTRACT

Multi-branch architectures are widely used in state-of-the-art neural networks. Their empirical success relies on some design wisdom, like adding normalization layers or/and scaling down the initialization. In this paper, we investigate the multi-branch architecture from the stability perspective. Specifically, we establish the forward/backward stability of multi-branch network, which leads to several new findings. Our analysis shows that only scaling down the initialization may not be enough for training multi-branch network successfully because of the uncontrollable backward process. We also unveil a new role of the normalization layer in terms of stabilizing the multi-branch architectures. More importantly, we propose a new design "STAM aggregation" that can guarantee to STAbilize the forward/backward process of Multi-branch networks irrespective of the number of branches. We demonstrate that with STAM aggregation, the same training strategy is applicable to models with different numbers of branches, which can reduce the hyper-parameter tuning burden. Our experiments verify our theoretical findings and also demonstrate that the STAM aggregation can improve the performance of multi-branch networks considerably.

## 1 INTRODUCTION

Multi-branch architecture is a building block in state-of-the-art neural network models for many tasks, e.g., the ResNeXt (Xie et al., 2017) for computer vision and the Transformer (Vaswani et al., 2017) for machine translation. It has been pointed out that the benefit of multi-branch architecture is the parameter efficiency (Xie et al., 2017). The number of parameters grows linearly with the number of branches but quadratically with the width (the number of neurons in one layer). It has also been argued that the multiple branches can bring diversity if branches are composed of sub-networks with different filters and depths (Huang et al., 2017; Li et al., 2019).

To train multi-branch networks successfully, it usually requires careful designs and some hyper-parameter tuning such as adding normalization layers, scaling down the initialization, and adjusting learning rate. As a verifying example, for a trainable single-branch network, simply adding branches multiple times and aggregating their outputs together often do not work as expected, e.g., the training instability of sum aggregation in Figure 4. This demonstrates the difficulty of training multi-branch network and also motivates us to do this study.

In this paper, we try to understand the behavior of training multi-branch network. Specifically, we study the forward and backward process of multi-branch networks, which is believed to govern whether the network is easy to optimize by gradient-based methods. We find out that the aggregation scheme, i.e., "the way of combining the multi-branch outputs" plays a central role in determining the behavior of training multi-branch network. We show that the sum aggregation would become unstable as the number of branches grows, which explains the bad performance of simply adding branches. Moreover, we characterize the condition on the aggregation scheme under which the forward and backward stability is guaranteed.

Inspired by the theoretical analysis, we propose a "STAM" aggregation, that can *STAbilize Multi-branch* network, which scales the sum of the branch outputs by a branch-aware factor $\alpha$ (see the later part of Section 3.1 for details). We argue the benefit of STAM aggregation over the sum and average aggregations by analyzing the Hessian of the multi-branch network. We show that STAM permits the same gradient-based optimizer works for different settings, which could reduce lots of tuning burden for training network with flexible number of branches.

We further examine the usual design wisdom through the stability lens. As a result, we find that scaling down initialization may control the forward or backward stability but not necessarily the both, which is verified in experiment. We also unveil a new role of normalization layer that it can stabilize the forward and backward process of multi-branch network besides the many wanted and unwanted properties that have been argued before (Ioffe & Szegedy, 2015; Yang et al., 2018; Santurkar et al., 2018; Xiong et al., 2020).

Apart from the usual feedforward multi-branch architecture, we analyze the multi-head attention layer, a multi-branch architecture widely used in natural language processing. We give an upper bound on the multi-head representations when the softmax operator is replaced with max operation. The upper bound unveils the relation between the head dimension and the length of the sequence, which interprets empirical observation well. This relation cannot be discovered if assuming softmax outputs equal probability as in Xiong et al. (2020).

Overall, our contribution can be summarized as follows.

- We analyze the forward/backward stability of multi-branch network, under which we can clearly interpret the benefit and potential problem of the practical wisdom, i.e., scaling down initialization and adding normalization layer.
- We propose a theoretically inspired STAM aggregation design for multi-branch network, which can handle arbitrary number of branches with a same optimizer.
- We also analyze the forward/backward process of multi-head attention layer and identify its special property that has not been characterized before.

## 1.1 RELATED WORK

Multi-branch architecture, also known as split-transform-merge architecture, has been widely used in computer vision task, namely Inceptions (Szegedy et al., 2017; Chollet, 2017), ResNeXt (Xie et al., 2017), and many others (Abdi & Nahavandi, 2016; Ahmed & Torresani, 2017). In fact, the models in natural language tasks have also leveraged the multi-branch architecture including the BiLSTM (Wu et al., 2016; Zhou et al., 2016) and the multi-head attention layer in Transformer (Vaswani et al., 2017; Anonymous, 2020). Apart from the sum or average aggregation, recent works (Li et al., 2019; Zhang et al., 2020) integrate the attention mechanism with the aggregation scheme, i.e., the attentive aggregation, although only a small number ($2 \sim 3$) of parallel branches are considered.

Theoretically, Zhang et al. (2018) interpret the benefit of multi-branch architecture from reducing the duality gap or the degree of non-convexity. The theory of training general deep neural network has been widely studied, via the stability analysis (Arpit et al., 2019; Zhang et al., 2019a;c; Yang & Schoenholz, 2017; Zhang et al., 2019b; Yang, 2019; Lee et al., 2019), neural tangent kernel (Jacot et al., 2018; Allen-Zhu et al., 2018; Du et al., 2018; Chizat & Bach, 2018; Zou et al., 2018; Zou & Gu, 2019; Arora et al., 2019; Oymak & Soltanolkotabi, 2019; Chen et al., 2019; Ji & Telgarsky, 2019). In contrast, we focus on the multi-branch network, which has not been studied theoretically before.

## 2 MODEL DESCRIPTION AND NOTATIONS

In practice, the multi-branch architecture is often used as a building block in a whole network. In this paper, we describe a multi-branch architecture/network $\mathcal{N}(\cdot)$ as follows (see Figure 1).

- $\mathcal{N}(\cdot)$ has $C$ branches $\{\mathcal{B}^k\}_{k=1}^C$, input $\boldsymbol{h}_{in} \in \mathbb{R}^p$ and output $\boldsymbol{h}_{out} \in \mathbb{R}^d$;
- The aggregation is parameterized with a vector $\boldsymbol{\alpha} = (\alpha_1, \ldots, \alpha_C)^T$:

$$\boldsymbol{h}_{out} := \mathcal{N}(\boldsymbol{h}_{in}) := \sum_{k=1}^C \alpha_k \cdot \mathcal{B}^k(\boldsymbol{h}_{in}). \tag{1}$$

Each branch $\mathcal{B}^k$ often consists of multiple layers with various structures and flexible configuration: depth, width, kernel size for convolution layer, activation functions and normalization layer. The aggregation weight is given by $\boldsymbol{\alpha}$. Such description covers popular multi-branch architectures in state-of-the-art models, e.g., Inception, ResNeXt and Transformer, if specifying $\mathcal{B}^k$ and $\boldsymbol{\alpha}$ properly.

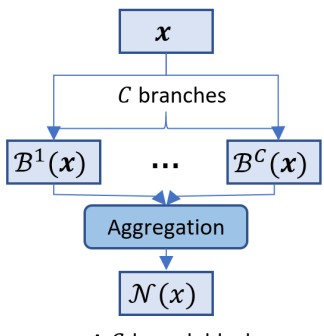

A $C$-branch block

Figure 1: A multi-branch network.

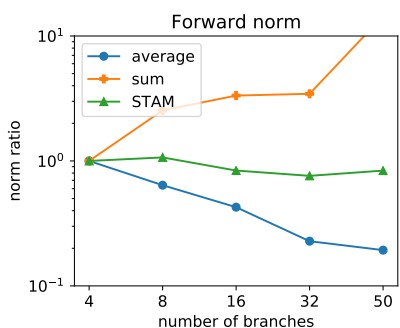

Figure 2: Output norm after first aggregation in multi-branch ResNets at initialization. The norm ratio is with respect to the case of $C = 4$.

Throughout the paper, we use $\| \cdot \|$ to denote the $l_2$ norm of a vector. We further use $\| \cdot \|$ and $\| \cdot \|_F$ to denote the spectral norm and the Frobenius norm of a matrix, respectively. We denote a set of naturals with $[n] := \{1, \ldots, n\}$ and $[c:d] = \{c, c+1, \ldots, d\}$. We use bold small case letters, e.g., $\boldsymbol{v}$, to denote vectors and bold capital case letters, e.g., $\boldsymbol{M}$, to denote matrices. Moreover, $\boldsymbol{I}_{d \times d}$ is the $d \times d$ identity matrix, $\boldsymbol{1}_d$ is $d$ dimensional vector with all 1's, and $\mathrm{Vec}(\boldsymbol{M})$ stacks the row vectors of matrix $\boldsymbol{M}$ as a long vector.

## 3 STABILITY OF MULTI-BRANCH FEEDFORWARD NETWORK

To theoretically study the multi-branch network, we introduce a simplified multi-branch feedforward network. Specifically, we assume that each branch is a $b$-layer fully connected network with ReLU activation, and each layer has the same width $m$. Branches share the same structure and they differ from each other by the random initialization. One branch is given by

$$\mathcal{B}^k(\boldsymbol{h}_{in}) = \boldsymbol{W}_b^k \phi(\boldsymbol{W}_{b-1}^k \cdots \phi(\boldsymbol{W}_1^k \boldsymbol{h}_{in})), \tag{2}$$

where $\boldsymbol{W}_1^k \in \mathbb{R}^{m \times p}, \boldsymbol{W}_b^k \in \mathbb{R}^{d \times m}$ and $\phi(\cdot)$ is the ReLU activation $\phi(\cdot) := \max\{0, \cdot\}$. We further introduce $\overrightarrow{\boldsymbol{W}^k} := (\boldsymbol{W}_1^k, \ldots, \boldsymbol{W}_b^k)$ that collects parameters of $\mathcal{B}^k$, and $\overrightarrow{\boldsymbol{W}} := (\overrightarrow{\boldsymbol{W}}^1, \ldots, \overrightarrow{\boldsymbol{W}}^C)$.

Next we analyze the forward and backward propagation of the multi-branch network given by (Equation 1) and (Equation 2), and characterize a condition that guarantees forward/backward stability. Based on the theoretical analysis, we propose the STAM aggregation that can stabilize the forward/backward process of multi-branch networks. We further argue why the practical wisdom of scaling down initialization and adding normalization layer works and when it could fail.

### 3.1 FORWARD AND BACKWARD PROCESS

We assume that the feedforward multi-branch network given by (Equation 1) and (Equation 2) adopts the Kaiming's initialization (He et al., 2016): entries of $\boldsymbol{W}_a$ for $a \in [1:b-1]$ are independently sampled from $\mathcal{N}(0, \frac{2}{m})$, and entries of $\boldsymbol{W}_b$ are independently sampled from $\mathcal{N}(0, \frac{1}{d})$. Then the forward norm is well concentrated around its mean value as follows.

**Theorem 1.** *Suppose the multi-branch network $\mathcal{N}(\cdot)$ is given by Equation 1 and Equation 2 with Kaiming's initialization. For an input $\boldsymbol{h}_{in}$, the following holds with probability at least $1 - O(bC) \cdot e^{-\Omega(m\epsilon^2/b)}$ over the initialization randomness of $\overrightarrow{\boldsymbol{W}}$*

$$\|\mathcal{N}(\boldsymbol{h}_{in})\| \in (1 \pm \epsilon)\sqrt{\sum_{k \in [C]} \alpha_k^2} \|\boldsymbol{h}_{in}\|, \tag{3}$$

*where $C$ is the number of branches, and $m, b$ are the width and depth of each branch, respectively.*

*Proof.* The proof is based on the Gaussianness of $\mathcal{B}^k(\boldsymbol{h}_{in})$ and a concentration property, whose full version is deferred to Appendix A.2. □

Theorem 1 is presented for one input sample. If we want such a result holds for all the training samples, the probability loses an $n$ factor by the union bound and becomes $1 - O(nbC) \cdot e^{-\Omega(m\epsilon^2/b)}$.

**Remark 1.** *With the same assumption as Theorem 1, we have*

$$\mathbb{E}\|\mathcal{N}(\boldsymbol{h}_{in})\| = \sqrt{\sum_{k \in [C]} \alpha_k^2} \|\boldsymbol{h}_{in}\|, \tag{4}$$

*where $C$ is the number of branches and the expectation is over the random initialization.*

These results are based on the bound of the forward propagation of one feed-forward branch given by (Equation 2), which is studied in Allen-Zhu et al. (2018) and restated in Appendix A. Furthermore, if the weight matrices follows the Gaussian distribution, then $\mathcal{B}^k(\boldsymbol{h}_{in})$ is also roughly Gaussian and jointly Gaussian for different input samples, as the width $m$ goes to infinity. Then the aggregation (Equation 1) can be viewed as a sum of weighted Gaussian vectors. Hence at the initialization, we can characterize the aggregation of multiple branches as above.

We next analyze the backward propagation of multi-branch feedforward network. We abuse "gradient" to refer to the values computed through back-propagation even for non-smooth function. We assume the loss function $\ell(\cdot, \cdot)$ is quadratic, i.e., $\ell(\boldsymbol{h}_{out}, \boldsymbol{y}^*) = \frac{1}{2}\|\boldsymbol{h}_{out} - \boldsymbol{y}^*\|_2^2$. Hence, the objective function is $\mathcal{L}(\overrightarrow{\boldsymbol{W}}) := \frac{1}{n}\sum_{i=1}^n \mathcal{L}_i(\overrightarrow{\boldsymbol{W}})$, where $\mathcal{L}_i(\overrightarrow{\boldsymbol{W}}) := \ell(\mathcal{N}(\boldsymbol{x}_i), \boldsymbol{y}_i^*)$. We next show the backward process is bounded for each individual sample for the multi-branch network $\mathcal{N}(\cdot)$.

**Theorem 2.** *With probability at least $1 - (nb) \cdot \exp(-\Omega(m))$ over the randomness of $\overrightarrow{\boldsymbol{W}}$, it satisfies for every $a \in [b]$ and $k \in [C]$, every $i \in [n]$,*

$$\left\|\nabla_{\boldsymbol{W}_a^k}\mathcal{L}_i(\overrightarrow{\boldsymbol{W}})\right\|_F^2 \leq O\left(\mathcal{L}_i(\overrightarrow{\boldsymbol{W}})\frac{\alpha_k^2 \times m}{d}\right), \quad \left\|\nabla_{\overrightarrow{\boldsymbol{w}}}\mathcal{L}_i(\overrightarrow{\boldsymbol{W}})\right\|_F^2 \leq O\left(\mathcal{L}_i(\overrightarrow{\boldsymbol{W}}) \times \frac{mb}{d}\sum_{k \in [C]}\alpha_k^2\right). \tag{5}$$

*Proof.* We can compute the gradient with respect to intermediate layer outputs via the backward propagation procedure. The gradient upper bound is guaranteed if the intermediate layer outputs and their gradients are bounded across layers. The full proof is relegated to Appendix A.3. □

If further assuming the gradient independence condition: weights in backward process can be assumed to be independent from weights in the forward pass (Yang, 2019), we can estimate the expectation of the gradient norm as follows.

**Remark 2.** *Assuming the gradient independence, we have for every $a \in [b]$ and $k \in [C]$, every $i \in [n]$,*

$$\mathbb{E}\left\|\nabla_{\boldsymbol{W}_a^k}\mathcal{L}_i(\overrightarrow{\boldsymbol{W}})\right\|_F^2 = \mathcal{L}_i(\overrightarrow{\boldsymbol{W}})\frac{\alpha_k^2 \times m}{d}, \quad \mathbb{E}\left\|\nabla_{\overrightarrow{\boldsymbol{w}}}\mathcal{L}_i(\overrightarrow{\boldsymbol{W}})\right\|_F^2 = \mathcal{L}_i(\overrightarrow{\boldsymbol{W}}) \times \frac{mb}{d}\sum_{k \in [C]}\alpha_k^2. \tag{6}$$

With Theorem 1 and 2 and two remarks, we can discuss the property of the forward and backward process of the multi-branch network. We can see that both the output of multi-branch network and the gradient are under control if $\sum \alpha_k^2 \leq O(1)$. Specifically, for the sum aggregation, we have $\sum \alpha_k^2 = C$ which grows unbounded with the number of branches $C$. For the average aggregation, we have $\sum \alpha_k^2 = 1/C$ which diminishes with the number of branches $C$.

There exists a better choice of $\alpha_k$: $\alpha_k = 1/\sqrt{C}$ for $k \in [C]$ that keeps $\sum \alpha_k^2 = 1$ constant as the number of branches varies. We call it "STAM" aggregation, abbreviating STAble Multi-branch aggregation. We plot the output norm of the first residual block in multi-branch ResNets at initialization in Figure 2. Multi-branch ResNets are generated by varying the number of branches in the residual block with batch normalization removed. We can see that the forward norm of STAM aggregation roughly remains the same, while that of the sum aggregation explodes and that of the average aggregation diminishes, as the number of branches grows.

We also analyze the Hessian of different aggregation schemes. We find that the spectral norm of the Hessian, which determines the smoothness of the objective, proportionally scales with the square root of the number of branches for the sum aggregation, while reciprocally scales with the square root of the number of branches for the average aggregation. In contrast, the Hessian for the STAM aggregation keeps unchanged as the number of branches varies. Hence with STAM, the same learning rate works for network with different number of branches. We present the details in Appendix B.

### 3.2 Understanding the Practical Wisdom

In practice people have design wisdom to stabilize the multi-branch network. We next analyze them through the stability lens.

One can *scale down the initialization* for multi-branch network. For example, one can initialize each $\boldsymbol{W}_l^k$ with a scaling-down factor $C^{-\frac{1}{2b}}$ for all $l \in [b]$ and $k \in [C]$ so that the forward norm of $\mathcal{N}(\boldsymbol{h}_{in})$ has a bound around $\|\boldsymbol{h}_{in}\|$, irrelevant with $C$. However, with this initialization, the norm of the output update induced by one gradient descent step, scales with $bC^{\frac{1}{b}}$. Alternatively, we suggest initializing each $\boldsymbol{W}_l^k$ with a scaling-down factor $(bC)^{\frac{-1}{2(b-1)}}$ for all $l \in [b]$ and $k \in [C]$ to stabilize the backward process such that the expected update on the output is irrelevant with $b$ and $C$. This indicates that a constant learning rate can train multi-branch network with any $b$ and $C$, although the forward process has a diminishing output norm scaling with $b^{-\frac{1}{2}}(bC)^{\frac{-1}{2(b-1)}}$. More detailed discussion is presented in Appendix A.4. We empirically verify this in Section 5.1.

Another widely used technique is adding normalization layers. It is easy to see that normalization layer can stabilize the forward process as it normalizes the output to be with mean $0$ and variance $1$. Moreover, the normalization layer can also stabilize the backward process as the error signal is divided by the standard deviation which is proportional to $\sqrt{C}$ when propagating backward. Therefore, we show a new role of the normalization layer that it automatically stabilizes the forward and backward process of the multi-branch network or in general stabilizes the architecture, besides the previous understanding of increasing the smoothness (Santurkar et al., 2018) or handling the covariance shift (Ioffe & Szegedy, 2015) or implicit structure bias (De & Smith, 2020).

We next examine concrete structures. One popular aggregation scheme for multi-branch outputs is concatenation followed by a linear transformation. This is equivalent to a sum aggregation. The default Xavier initialization (Glorot & Bengio, 2010) would scale down the initialization of the linear transformation by roughly $1/\sqrt{C}$ to stabilize the forward process but the training could still be unstable because of the unbounded backward process as discussed above and verified in Section 5.1.

A well-known multi-branch structure is ResNeXt for image classification task. It uses batch normalization on the output of the residual branch to stabilize the forward and backward processes. However, if we use the pre-act form, i.e., remove the batch normalization on the output of the residual branch and add a batch normalization on the input of the residual branch, the training becomes unstable as the number of branches increases (see Section 5.2). With the pre-act residual block, one needs a normalization layer on the final output of the network, which has been studied in (Xiong et al., 2020). It could still be unstable for the deep half-precision ResNet, as the output may overflow before the final normalization layer. This is interesting but beyond the scope of this paper.

Transformer also uses multi-branch structure in the attention layer and fully-connected layer. It does not use normalization layer on the output of the residual block because of the practical performance concerns. Hence it is observed unstable training or deteriorating performance when adding the number of heads in attention layer or adding fully-connected branches (see Section 5.3).

## 4 Forward and Backward Process of Multi-head Attention Layer

In this section, we analyze the multi-head attention layer, a multi-branch structure used in Transformer (Vaswani et al., 2017) with each head viewed as one branch. It is worthy to note that the multi-head attention layer behaves differently from the feed-forward network and at the same time it is in general very hard to analyze, because of the softmax operator and the inter-symbol dependence. Previous work assumes that the softmax outputs equal weights Xiong et al. (2020) and ignores the inter-symbol dependence, which does not fully reflect the attention behavior.

Suppose that the input is a sequence of symbols $\boldsymbol{X} = (\boldsymbol{x}_1, \ldots, \boldsymbol{x}_n)^T$ and each row of $\boldsymbol{X}$ is a symbol representation $\boldsymbol{x}_i \in \mathbb{R}^d$. The multi-head self-attention layer, is given by

$$\text{MultiHead}(\boldsymbol{Q}, \boldsymbol{K}, \boldsymbol{V}) = \text{Concat}(\boldsymbol{h}_1, ..., \boldsymbol{h}_C)\tilde{\boldsymbol{O}}, \tag{7}$$

$$\boldsymbol{h}_c = \text{softmax}\left(\frac{\boldsymbol{Q}\boldsymbol{K}^T}{\sqrt{d_k}}\right)\boldsymbol{V}, \quad \text{for } c \in [C], \tag{8}$$

where $Q := X\tilde{Q}_c, K := X\tilde{K}_c, V := X\tilde{V}_c$ and $\tilde{Q}_c \in \mathbb{R}^{d \times d_k}, \tilde{K}_c \in \mathbb{R}^{d \times d_k}, \tilde{V}_c \in \mathbb{R}^{d \times d_v}, \tilde{O}_c \in \mathbb{R}^{d_v \times d}$ are projection (trainable) matrices and $C$ is the number of heads, $d_v, d_k$ are dimensions.

The multi-head attention computes each head representation (with dimension $d_v$) independently, and then concatenates these heads ($C$ in total) together to form a large representation ($Cd_v$ dimensional vector), finally uses a fully-connected layer $\tilde{O} \in \mathbb{R}^{d \times (Cd_v)}$ projecting back to $\mathbb{R}^d$ ($d$ is the model embedding dimension). This indicates that the original multi-head attention uses the sum aggregation. It is convenient to first consider the case of one head with dimension $d_v$ for studying the multi-head attention layer. We give an upper bound on the forward process when the softmax behaves more like a "max" operator. The softmax output could be rather spreadout even after training which is usually the case after initial training.

**Proposition 1.** *Suppose a self-attention layer as described above and assume $\|x_i\| = \sqrt{d}$ for $i \in [n]$. If the parameters $\tilde{Q}, \tilde{K}, \tilde{V}$ are randomly initialized with $\mathcal{N}(0, \frac{1}{d})$, then for symbol $x_i$ and its head representation $h_i \in \mathbb{R}^{d_v}$, we have*

$$\mathbb{E}\|h_i\|^2 \lesssim d_v + (\log n - 1 + \sqrt{2(2d_v - 1)\log n}), \tag{9}$$

*where "$\lesssim$" means that the bound holds asymptotically and the expectation is over the initialization.*

*Proof.* The proof is based on estimating the extreme Chi-square values, deferred to Appendix C. $\square$

We can see that besides $d_v$, the norm also scales with $\log n$ where $n$ is the number of symbols. In the Transformer literature, one usually fixes $C \cdot d_v = d$, and then the upper bound of head norm grows with $C$ because of the second term in the right hand side of Equation 9. This term is ignored in previous analysis, which can dominate when $d_v < \Omega(\log n)$. We plot the norms of self attention layer (in the first decoder block) in Transformers with varying number of heads on IWSLT de-en task in Figure 3 to illustrate this point. We can see at initialization the norm does not change with number of heads because of the expected behavior at initialization (Xiong et al., 2020). However, this is not true after training (see the curve at Epoch 10). The assumption that softmax outputs uniform simplex does not hold. With a large $C$ and a small $d_v$, the second term in Equation 9 dominates, which may hurts the forward/backward stability and the training process, matching the observation in practice.

For the backward process, we can estimate the gradient at initialization with the gradient independence assumption (Yang, 2019). Suppose the backward signal is $\{e_1, e_2, ..., e_n\}$, the gradient on $\tilde{O}$ is $\nabla_{\tilde{O}} = \sum_{i \in [n]} e_i h_i^T$ and $\mathbb{E}\|\nabla_{\tilde{O}}\|_F^2 = \sum_{i \in [n]} (\mathbb{E}\|h_i\|^2)\|e_i\|_2^2$, which scales with the head norm. Large forward process results in large gradients, which may lead to unstable training procedure.

## 5 EXPERIMENTS

In this section, we first verify the theoretical claims in Section 3. We then apply the STAM aggregation on the popular ResNeXt (Xie et al., 2017) model for image classification and on Transformer (Vaswani et al., 2017) to demonstrate its efficacy.

### 5.1 THEORETICAL VERIFICATION

We conduct some experiments to verify the theoretical claims in Section 3. Specifically, we use a Fixup ResNet20 (Zhang et al., 2019a) as a backbone network, which uses Fixup initialization without all batch normalization (BN) layers and can be trained to a decent accuracy. The multi-branch ResNets are generated by varying the number of branches in each residual block (Figure 1). We train these networks on the CIFAR10/CIFAR100 classification task (Krizhevsky & Hinton, 2009) and record the top 1 validation error in Figure 4 with a standard training procedure, i.e., 128 batch size, 200 epoch, initial learning rate 0.1 and step-wise learning rate decay by $1/10$ at epoch 100 and 150 respectively.

We compare several approaches: the STAM aggregation with $\alpha = 1/\sqrt{C}$; the sum aggregation with $\alpha = 1$; the average aggregation with $\alpha = 1/C$; the backward scaling-down initialization $\gamma = 1/\sqrt{C}$, referred to as "Back-scaled init" in Figure 4; the concatenation then aggregation by a linear layer with scaling-down initialization $\gamma = 1/\sqrt{C}$, referred to as "linear layer aggregation" in Figure 4. As

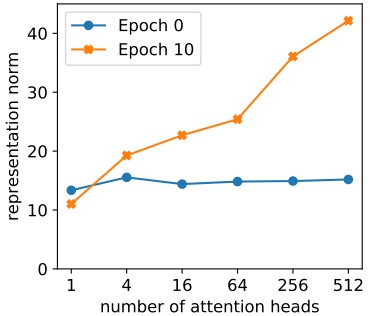
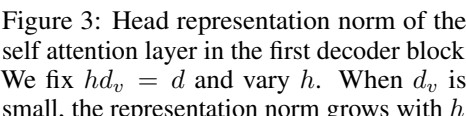

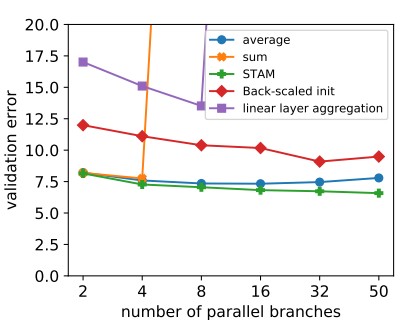

Figure 3: Head representation norm of the self attention layer in the first decoder block. We fix $hd_v = d$ and vary $h$. When $d_v$ is small, the representation norm grows with $h$.

Figure 4: Top1 validation error of multi-branch ResNets on CIFAR10. Models trained with STAM aggregation consistently benefit from increasing branches.

Table 1: Top1 validation accuracy (in %) of ResNeXt models. The numbers with * are reported in (Xie et al., 2017) and other numbers are our implementation (averaged over 3 repeats).

| Dataset | Setting | Param. | Basline | +STAM |
|---|---|---|---|---|
| CIFAR10 | 8×64d | 34.4M | 96.35* | **96.51**(0.07) |
| | 16×64d | 68.1M | 96.42* | **96.57**(0.12) |
| | 32×64d | 135.9M | 96.37(0.11) | **96.73**(0.13) |
| CIFAR100 | 8×64d | 34.4M | 82.23* | **83.17**(0.11) |
| | 16×64d | 68.1M | 82.69* | **83.63**(0.09) |
| | 32×64d | 135.9M | 83.26(0.14) | **83.86**(0.12) |

shown in Figure 4, the performance of model trained with STAM aggregation consistently improves as the number of branches $C$ grows. In contrast, the models with sum aggregation fail to converge for $C > 4$, as predicted in Theorem 1 and 2. The performance of average aggregation degrades as the number of branches is beyond a certain value. In addition, the backward scaling-down initialization works well in terms of stability as we argued in Section 3.2. However, the scheme of concatenation then aggregation with a linear transformation becomes unstable as the number of branches increases even with a scaled-down initialization.

## 5.2 APPLY STAM AGGREGATION TO TRAIN RESNEXT

We next investigate a well-known multi-branch network, the ResNeXt (Xie et al., 2017) model, which divides the bottleneck residual block into multiple branches. It uses 1x1 convolution to aggregate the branch outputs, which is equivalent to sum aggregation if taking the 1x1 convolution as part of each branch (see Figure 3 in Xie et al. (2017)). ResNext uses a BN layer to normalize the aggregated result. We have argued how normalization layer can stabilize the forward and backward process in Section 3. It can be observed that ResNeXt fails to converge for $C > 2$ without such BN layer.

To verify the efficacy of STAM aggregation on ResNeXt, we use separate BN layer for each branch, equivalent to viewing the BN layer a part of a branch, and then apply the STAM aggregation on these outputs. The hyperparameters and the data augmentation are the same as in Xie et al. (2017). From Table 1, we see that our STAM-aggregation is applicable to the cases with many branches (more than the original paper) and improves over the strong baselines on both CIFAR10/CIFAR100. More details about the experiments can be found in Appendix D.

## 5.3 APPLY STAM AGGREGATION TO TRAIN TRANSFORMER WITH MANY BRANCHES

In this section, we investigate the aggregation scheme on Transformer models (Vaswani et al., 2017) with machine translation tasks. A Transformer model consists of several encoder and decoder layers, with each layer stacking one or two multi-head attention blocks and one fully connected (FC) block.

Table 2: BLEU scores on IWSLT14 de-en test sets. The higher, the better. * is baseline setting.

| $h/d/d_{fc}$ | Param. | BLEU | BLEU (+STAM) |
|---|---|---|---|
| 4/512/1024 | 36.7M | 35.21* | N/A |
| 8/256/1024 | 18.4M | 34.96 | 35.30 |
| 12/256/1536 | 26.3M | 35.0 | 35.79 |
| 16/256/2048 | 34.2M | 34.74 | 36.05 |
| 32/256/4096 | 65.7M | 34.30 | **36.09** |

Table 3: BLEU scores on newstest2014 for WMT En-De. The higher, the better. The number with * is reported in (Ott et al., 2018).

| $h/d/d_{fc}$ | Param. | BLEU | BLEU (+STAM) |
|---|---|---|---|
| 16/1024/4096 | 209.8M | 29.3* | N/A |
| 16/512/4096 | 105.1M | 28.6 | 28.8 |
| 32/512/8192 | 193.2M | 24.3 | 29.4 |
| 36/512/9216 | 215.2M | 23.5 | **29.8** |

There are several hyperparameters about the model setting: model/embedding dimension $d$, head dimension $d_h$, number of heads $h$, intermediate FC dimension $d_{fc}$. More detailed introduction on Transformer can be found in Appendix C.

We conduct two sets of experiments: IWSLT14 German-English (de-en) task and the WMT16 English-German (en-de) task. For baseline models, we choose the default Transformer and the big Transformer for IWSLT14 de-en and WMT16 en-de, respectively. For multi-branch versions of Transformer, we first construct a mini model, and then generate multi-branch Transformers by copying mini-models multiple times. This can explore many branches while controlling model size.

For IWSLT14 de-en task, the baseline Transformer configuration is $d = 512, d_h = 128, d_{fc} = 1024, h = 4$. We choose the mini model for IWSLT14 de-en with configuration $d = 256, d_h = 64, h = 4, d_{fc} = 512$, and the generated multi-branch Transformers are with configurations $(d, d_h, h, d_{fc}) = (256, 64, 8, 1024), (256, 64, 12, 1536), ...$ . These models are trained with the Adam (Kingma & Ba, 2014) optimizer with initial learning rate 1e-3, $\beta_1 = 0.9, \beta_2 = 0.98$ and the *inverse sqrt* learning rate scheduler, which are standard hyperparemeter choices following Ott et al. (2018). For the STAM aggregation, we set $\alpha = 1/\sqrt{C}$ as $C$ is the equivalent number of mini models. We evaluate the translation quality by BLEU scores by using *multi-bleu.perl*. We train every model on a single Tesla P40 GPU for 200 epochs. We test on a single model that achieves best BLEU score on the validation set. Other detailed configuration can be found in the code.

For the WMT16 English-German (en-de) task, the baseline big Transformer model has configuration $d = 1024, d_h = 64, d_{fc} = 4096, h = 16$. We choose the mini model for WMT16 en-de task with configuration $d = 512, d_h = 64, h = 8, d_{fc} = 2048$, and the generated multi-branch Transformers are with configurations $(d, d_h, h, d_{fc},) = (512, 64, 16, 4096), (512, 64, 32, 8192), ...$ . For the STAM aggregation, we set $\alpha = 1/\sqrt{C}$ where $C$ is the equivalent number of mini models. All models are trained for 150 epochs on a single node with 8 Tesla V100 GPUs. We average the model parameters of last 5 checkpoints for evaluation.

We report the BLEU scores of all the models on test set in Table 2 and 3. We can see that the performance of original architecture degrades as the number of branches increases, while with STAM aggregation the performance gradually improves as the number of branches increases.

# 6 CONCLUSION

In this paper, we study the training process of multi-branch network, especially the forward and backward stability. The theory tells that the sum aggregation is not stable as the number of branches increases. Motivated by the theoretical analysis of the multi-branch network, we propose the STAM aggregation that can not only guarantee the forward/backward stability but also allows a same training strategy works for networks with different number of branches. We show that with STAM aggregation, the models can consistently benefit from increasing the number of branches. We believe that our analysis and the proposed STAM aggregation gives practitioners a new direction to design new multi-branch models.

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

# A PROOFS IN SECTION 3

For feed-forward branches given by equation 2, we adopt the Kaiming's initialization He et al. (2016): entries of $\boldsymbol{W}_a$ for $a \in [1 : b - 1]$ are independently sampled from $\mathcal{N}(0, \frac{2}{m})$, and entries of $\boldsymbol{W}_b$ are independently sampled from $\mathcal{N}(0, \frac{1}{d})$.

## A.1 ONE BRANCH RESULT

**Lemma 1.** *[Lemma 7.1 in Allen-Zhu et al. (2018)] For one branch $\mathcal{B}(h_{in})$ given by equation 2 with Kaiming's initialization, there exists some small constant $\epsilon$ such that with probability at least $1 - O(b) \cdot e^{-\Omega(m\epsilon^2/b)}$ over the initialization randomness of $\overrightarrow{\boldsymbol{W}} \in (\mathbb{R}^{m \times m})^b$ the following holds*

$$\forall a \in [b-1] : \|h_a\| \in (1 \pm \epsilon)\|h_{in}\|, \tag{10}$$

*where $h_a := \phi(\boldsymbol{W}_a h_{a-1})$, for $a = 1, ..., b-1$ with $h_0 := h_{in}$.*

We note that $\|h_a\| \in (1 \pm \epsilon)\|h_{in}\|$ means $(1 - \epsilon)\|h_{in}\| \leq \|h_a\| \leq (1 + \epsilon)\|h_{in}\|$. The proof of this lemma is based on the randomness of $\overrightarrow{\boldsymbol{W}}^k$ and the concentration property.

## A.2 PROOF OF THEOREM 1

**Theorem 1.** *Suppose the multi-branch network $\mathcal{N}(\cdot)$ is given by Equation 1 and Equation 2 with Kaiming's initialization. For an input $\boldsymbol{h}_{in}$, the following holds with probability at least $1 - O(bC) \cdot e^{-\Omega(m\epsilon^2/b)}$ over the initialization randomness of $\overrightarrow{\boldsymbol{W}}$*

$$\|\mathcal{N}(\boldsymbol{h}_{in})\| \in (1 \pm \epsilon)\sqrt{\sum_{k \in [C]} \alpha_k^2}\|\boldsymbol{h}_{in}\|, \tag{3}$$

*where $C$ is the number of branches, and $m, b$ are the width and depth of each branch, respectively.*

*Proof.* In the following derivation, we fix a specific sample $i \in [n]$ and the input is $h_{i,in}$. For notation simplicity, we ignore the $i$ index. We define an event $\mathcal{E} := \{\|h_{b-1}^k\| \in (1 \pm \epsilon)\|h_{in}\|, \forall k \in [C]\}$. By Lemma 1, we have that $\mathcal{E}$ holds with probability $1 - O(bC)e^{-\Omega(m\epsilon^2/b)}$. On the event $\mathcal{E}$, using only the randomness of $\boldsymbol{W}_b^k \in \mathbb{R}^{d \times m}$, $h_b^k = \boldsymbol{W}_b^k h_{b-1}^k$ are independent Gaussian across $k$ and $\mathbb{E}[h_b^k] = 0, \text{Var}[h_b^k] = \|h_{b-1}^k\|^2 \boldsymbol{I}_{d \times d}$ for $k \in [C]$.

Hence, we have $\mathcal{N}(h_{in}) = \sum_{k \in [C]} \mathcal{B}^k(h_{in})$ is Gaussian with mean 0 and covariance matrix $\sum_{k \in [C]} \alpha_k^2 \|h_{b-1}^k\|^2 \boldsymbol{I}_{d \times d}$. By law of large number, we have

$$\|\mathcal{N}(h_{in})\|^2 \in (1 \pm \epsilon)^2 \sum_k \alpha_k^2 \|h_{in}\|^2. \tag{11}$$

The claim is proved. $\square$

**Proof of Remark 1**

The proof of Remark 1 is straightforward. In expectation we always have $\mathbb{E}[h_l^k | h_{l-1}^k] = 0, \mathbb{E}\left[\|h_l^k\|^2 \big| h_{l-1}^k\right] = \|h_{l-1}^k\|^2$ for $k \in [C]$ and $l \in [b]$. Furthermore as $h_b^k$'s are independent across $k$, then $\mathbb{E}\|\mathcal{N}(h_{in})\|^2 = \sum_k \alpha_k^2 \|h_{in}\|^2$.

## A.3 PROOF OF THEOREM 2

Given the branch equation 2, we further introduce notations to denote intermediate representation:

$$g_a^k := \boldsymbol{W}_a^k h_{a-1}^k, \text{ and } h_a^k := \phi(g_a^k), \text{ for } a = 1, ..., b.$$

with $h_0^k := h_{in}$ and $h_b^k := g_b^k$ for all $k \in [C]$. Let $\boldsymbol{D}_a^k$ be a diagonal matrix representing the activation state with $(\boldsymbol{D}_a^k)_{jj} = 1$, if $(g_a^k)_j > 0; 0$, otherwise. Let $\overrightarrow{g_b^k} := (g_b^1, g_b^2, \ldots, g_b^k) \in \mathbb{R}^{d \times b}$.

**Theorem 2.** *With probability at least $1 - (nb) \cdot \exp(-\Omega(m))$ over the randomness of $\overrightarrow{\boldsymbol{W}}$, it satisfies for every $a \in [b]$ and $k \in [C]$, every $i \in [n]$,*

$$\left\| \nabla_{\boldsymbol{W}_a^k} \mathcal{L}_i(\overrightarrow{\boldsymbol{W}}) \right\|_F^2 \leq O\left( \mathcal{L}_i(\overrightarrow{\boldsymbol{W}}) \frac{\alpha_k^2 \times m}{d} \right), \quad \left\| \nabla_{\overrightarrow{\boldsymbol{w}}} \mathcal{L}_i(\overrightarrow{\boldsymbol{W}}) \right\|_F^2 \leq O\left( \mathcal{L}_i(\overrightarrow{\boldsymbol{W}}) \times \frac{mb}{d} \sum_{k \in [C]} \alpha_k^2 \right). \quad (5)$$

Deep neural networks are often trained with gradient based optimization algorithms. The gradient with respect to the parameter is computed through back-propagation, e.g., $\partial \boldsymbol{W}_l = \partial h_l \cdot g_{l-1}^T$, where $\partial \cdot$ represents the gradient of the objective with respect to $\cdot$. Therefore, the gradient upper bound is guaranteed if $g_{l-1}$ and $\partial h_l$ are bounded across layers and iterations. We next show the backward process is bounded for each individual sample for the multi-branch network $\mathcal{N}(\cdot)$.

*Proof.* The gradient is computed as follows via back-propagation. We ignore the sample index $i$ for notational simplicity.

$$h_{out} = \overrightarrow{g_b} \boldsymbol{\alpha}, \quad \frac{\partial h_{out}}{\partial g_b^k} = \alpha_k \boldsymbol{I}_{d \times d}, \quad (12)$$

$$\partial g_b^k = \left( \frac{\partial h_{out}}{\partial g_b^k} \right)^T \cdot \partial h_{out} = \alpha_k \partial h_{out}, \quad (13)$$

$$\partial h_{b-1}^k = (\boldsymbol{W}_b^k)^T \partial g_b^k, \quad \partial \boldsymbol{W}_b^k = \partial g_b^k \cdot (h_{b-1}^k)^T \quad (14)$$

$$\partial g_{b-1}^k = \boldsymbol{D}_{b-1}^k \partial h_{b-1}, \cdots \quad (15)$$

$$\partial h_0^k = (\boldsymbol{W}_1^k)^T \partial g_1^k, \quad \partial \boldsymbol{W}_1^k = \partial g_1^k \cdot (h_0^k)^T. \quad (16)$$

For quadratic loss and sample $i \in [n]$, we have that $\|\partial h_{i,out}\|^2 = 2\mathcal{L}_i(\overrightarrow{\boldsymbol{W}})$.

We have

$$\begin{aligned} \|\nabla_{\boldsymbol{W}_a^k} \mathcal{L}_i(\overrightarrow{\boldsymbol{W}})\|_F &= \|\partial g_{i,a}^k \cdot h_{i,a-1}^k\|_F \\ &= \|\boldsymbol{D}_a^k(\boldsymbol{W}_{a+1}^k)^T \cdots \boldsymbol{D}_{b-1}^k(\boldsymbol{W}_b^k)^T \partial h_{i,out}\| \cdot \|h_{i,a-1}^k\| \\ &\leq O(\sqrt{m/d}) \cdot \alpha_k \sqrt{\mathcal{L}_i(\overrightarrow{\boldsymbol{W}})} \cdot O(1)\|x_i\|, \end{aligned} \quad (17)$$

where the last inequality is due to Theorem 1 and the (Allen-Zhu et al., 2018, Lemma7.4b).

Thus, with high probability $1 - O(nbC) \cdot \exp(-\Omega(m))$ for all $i \in [n]$, $a \in [b]$ and $k \in [C]$, we have $\|\nabla_{\boldsymbol{W}_a^k} \mathcal{L}_i(\overrightarrow{\boldsymbol{W}})\|_F^2 \leq O\left( \frac{\mathcal{L}_i(\overrightarrow{\boldsymbol{W}})}{d} \alpha_k^2 \times m \right)$ under the assumption $\|x_i\| = 1$. $\qquad \square$

**Proof of Remark 2**

To establish the expectation value on the backward process, it requires the gradient independence assumption (Yang, 2019), which can be verified and argued for certain cases. With this assumption, we can take the expectation on the forward pass and the backward pass independently in (Equation 17). It is easy to obtain the expectation estimation in Remark 2.

### A.4    MORE DISCUSSION ON PRACTICAL WISDOM

We argue the choices of scaling down initialization in detail.

Without loss of generality, we first initialize the parameters following the Kaiming's scheme, the same as in Section 3. Then we multiply the parameters in multi-branch block by a same scaling down factor $\gamma$. We use the sum aggregation $\alpha_k = 1$ for $k \in [C]$.

For forward stability, we need $\mathbb{E}\|\mathcal{N}(h_{in})\|^2 \leq O(1)$, which requires $\sum_{k \in [C]} (\gamma^b)^2 \leq 1$. Therefore we obtain $\gamma \leq C^{-\frac{1}{2b}}$.

For the backward process, we focus on the output update induced by one gradient descent step. That is $\mathcal{N}(h_{in}; \overrightarrow{\boldsymbol{W}} + \eta \nabla_{\overrightarrow{\boldsymbol{W}}}) - \mathcal{N}(h_{in}; \overrightarrow{\boldsymbol{W}})$. Suppose the backward error signal on $h_{out}$ is $\boldsymbol{e}$ with $\|\boldsymbol{e}\| = 1$.

Then following the proof of Theorem 2, we have

$$\mathbb{E}\|\nabla_{\boldsymbol{W}_a^k}\|_F = \gamma^{b-1} \cdot \sqrt{m/d} \cdot \|\boldsymbol{e}\| \cdot \|h_{in}\| = \Omega(\gamma^{b-1}), \tag{18}$$

$$\mathbb{E}\|\mathcal{B}^k(h_{in}; \boldsymbol{W}_l^k + \eta\nabla_{\boldsymbol{W}_l^k}) - \mathcal{B}^k(h_{in}; \boldsymbol{W}_l^k)\| = \Omega(\eta\gamma^{2(b-1)}), \tag{19}$$

where the second equality is due to Taylor expansion and the forward step has a factor $\gamma^{b-1}$. Then the forward output update of branch $k$ is

$$\mathcal{B}^k\left(h_{in}; \overrightarrow{\boldsymbol{W}}^k + \eta\nabla_{\overrightarrow{\boldsymbol{W}}^k}\right) - \mathcal{B}^k\left(h_{in}; \overrightarrow{\boldsymbol{W}}^k\right) \approx \sum_{l\in[b]} \mathcal{B}^k\left(h_{in}; \boldsymbol{W}_l^k + \eta\nabla_{\boldsymbol{W}_l^k}\right) - \mathcal{B}^k\left(h_{in}; \boldsymbol{W}_l^k\right),$$

where the "$\approx$" ignores second order perturbation. Hence $\mathbb{E}\|\mathcal{B}^k(h_{in}; \overrightarrow{\boldsymbol{W}}^k + \eta\nabla_{\overrightarrow{\boldsymbol{W}}^k}) - \mathcal{B}^k(h_{in}; \overrightarrow{\boldsymbol{W}}^k)\| \approx \Omega(\eta b\gamma^{2(b-1)})$. Summing over $C$ branches, we have

$$\mathbb{E}\|\mathcal{N}(h_{in}; \overrightarrow{\boldsymbol{W}} + \eta\nabla_{\overrightarrow{\boldsymbol{W}}}) - \mathcal{N}(h_{in}; \overrightarrow{\boldsymbol{W}})\| \approx \Omega(\eta bC\gamma^{2(b-1)}). \tag{20}$$

Hence, to make the expected output update irrelevant with $C$ and $b$, it requires $\gamma = (bC)^{-\frac{1}{2(b-1)}}$.

With $\gamma = (bC)^{-\frac{1}{2(b-1)}}$, the forward process will not explode but obtain a diminishing output norm $\Omega(b^{-\frac{1}{2}}(bC)^{\frac{-1}{2(b-1)}})$.

## B  HESSIAN ANALYSIS

Next to obtain a refined understanding on the property of different aggregation schemes, we analyze the Hessian of a multi-branch network. Specifically for simplicity, we assume that $\mathcal{B}^k$ has only one linear layer: $\mathcal{B}^k(h_{in}) = \boldsymbol{W}^k h_{in}$. We can compute the Hessian of the objective with respect to the input $h_{in}$ and the learnable parameter $\overrightarrow{\boldsymbol{W}}$,

$$\boldsymbol{H}_{h_{in}} = \big(\sum_k \alpha_k \boldsymbol{W}^k\big)^T \big(\sum_k \alpha_k \boldsymbol{W}^k\big), \ \ \boldsymbol{H}_{\overrightarrow{\boldsymbol{W}}} = \big(\boldsymbol{\alpha}\boldsymbol{\alpha}^T\big) \otimes \boldsymbol{I}_{d\times d} \otimes \big(\mathbb{E}h_{in}h_{in}^T\big), \tag{21}$$

where $\boldsymbol{H}_{\overrightarrow{\boldsymbol{W}}} := \boldsymbol{H}_{\text{Vec}(\overrightarrow{\boldsymbol{W}})}$ and $\text{Vec}(\overrightarrow{\boldsymbol{W}}) := (\text{Vec}(\boldsymbol{W}^1)^T, \text{Vec}(\boldsymbol{W}^2)^T, \ldots, \text{Vec}(\boldsymbol{W}^k)^T)^T$ is the long vector stacking $\overrightarrow{\boldsymbol{W}}$, $\otimes$ is the Kronecker product, and $\mathbb{E}$ is average over the training samples.

**Fact 1.** *For $\mathcal{B}^k(h_{in}) = \boldsymbol{W}^k h_{in}$ and $\boldsymbol{\alpha} = \alpha\mathbf{1}_C$, we have the spectral norm of Hessian,*

$$\mathbb{E}\|\boldsymbol{H}_{h_{in}}\| = \Omega(\alpha^2 C), \ \ and \ \ \|\boldsymbol{H}_{\overrightarrow{\boldsymbol{W}}}\| = \alpha^2 C\|\mathbb{E}h_{in}h_{in}^T\|. \tag{22}$$

*where the first expectation is over the randomness at initialization and the second expectation is over the training samples.*

*Proof.* The Hessians are written as

$$\boldsymbol{H}_{h_{in}} = \big(\sum_k \alpha_k \boldsymbol{W}^k\big)^T \big(\sum_k \alpha_k \boldsymbol{W}^k\big) \overset{\alpha_k=\alpha}{=} \alpha^2 \Big(\sum_k \boldsymbol{W}^k\Big)^T \Big(\sum_k \boldsymbol{W}^k\Big),$$

$$\boldsymbol{H}_{\overrightarrow{\boldsymbol{W}}} = \big(\boldsymbol{\alpha}\boldsymbol{\alpha}^T\big) \otimes \boldsymbol{I}_{d\times d} \otimes \big(\mathbb{E}h_{in}h_{in}^T\big) \overset{\alpha_k=\alpha}{=} \alpha^2 \big(\mathbf{1}_{C\times C}\big) \otimes \big(\boldsymbol{I}_{b\times b} \otimes \big(\mathbb{E}h_{in}h_{in}^T\big)\big),$$

where $\otimes$ represents the Kronecker product and $\boldsymbol{H}_{\overrightarrow{\boldsymbol{W}}}$ has dimension $Cbp \times Cbp$.

Because entries of $\boldsymbol{W}^k$ are initialized with Gaussian distribution $\mathcal{N}(0, 1/d)$, $\sum_k \boldsymbol{W}^k$ follows Gaussian distribution $\mathcal{N}(0, C/d)$. The singular value of Gaussian matrix is given by the following Bai-Yin's law (BAI & YIN, 1993).

**Lemma 2.** *Let $\boldsymbol{A} \in \mathbb{R}^{N\times n}$, and entries of $\boldsymbol{A}$ are independent standard Gaussian random variables. Suppose that dimensions $N$ and $n$ grow to infinity while the aspect ratio $n/N$ converges to a constant in $[0, 1]$, one has*

$$s_{\max}(\boldsymbol{A}) = \sqrt{N} + \sqrt{n} + o(\sqrt{n}) \quad almost \ surely. \tag{23}$$

*where $s_{\max}(\boldsymbol{A})$ are the largest singular value of $\boldsymbol{A}$.*

Hence $\mathbb{E}\|\boldsymbol{H}_{h_{in}}\| = \alpha^2 C(1 + \sqrt{m/d})^2 = \Omega(\alpha^2 C)$ if viewing $m/d$ some fixed hyper-parameter.

For the spectral norm of $H_{\overrightarrow{\boldsymbol{W}}}$, we use the property of Kronecker product $\lambda_{\max}(\boldsymbol{A} \otimes \boldsymbol{B}) = \lambda_{\max}(\boldsymbol{A}) \cdot \lambda_{\max}(\boldsymbol{B})$, where $\lambda_{\max}(\cdot)$ is the largest eigenvalue of the positively semi-definite matrix $\cdot$. Hence

$$\|H_{\overrightarrow{\boldsymbol{W}}}\| = \alpha^2 \lambda_{\max}(\mathbf{1}_{C \times C}) \cdot 1 \cdot \lambda_{\max}(\mathbb{E} h_{in} h_{in}^T) = \alpha^2 C \|\mathbb{E} h_{in} h_{in}^T\|, \tag{24}$$

where the second equality is due to the fact $\lambda_{\max}(\mathbf{1}_{C \times C}) = C$ and $\lambda_{\max}(\boldsymbol{I}_{C \times C}) = 1$. $\qquad\square$

The spectral norm of the Hessian corresponds to the smoothness or the Lipschitz coefficient of the gradient. It determines the largest allowable learning rate for gradient descent algorithm. For the sum aggregation, the spectral norm $\|\boldsymbol{H}_{h_{in}}\| = \Omega(C)$ and $\|\boldsymbol{H}_{\overrightarrow{\boldsymbol{W}}}\| = C\|\mathbb{E} h_{in} h_{in}^T\|$. Hence they both are $C$ times larger than those in the case with only one branch. This indicates the learning rate for the gradient descent needs properly scaling down with the number of branches to guarantee convergence. For the average aggregation, at initialization, the spectral norm $\|\boldsymbol{H}_{h_{in}}\| = \Omega(1/C)$ and $\|\boldsymbol{H}_{\overrightarrow{\boldsymbol{W}}}\| = \frac{1}{C}\|\mathbb{E} h_{in} h_{in}^T\|$. They both are shrunk by $1/C$ compared to the case with only one branch. This indicates the learning rate or the gradient needs properly scaling up with the number of branches. Thus one cannot apply the same training strategy to train network with varying number of branches and the sum or average aggregation.

In contrast, for the STAM aggregation, the spectral norm $\|\boldsymbol{H}_{h_{in}}\| = \Omega(1)$ and $\|\boldsymbol{H}_{\overrightarrow{\boldsymbol{W}}}\| = \|\mathbb{E} h_{in} h_{in}^T\|$. They both remain the same when the number of branches varies. This indicates that we can apply the same training strategy for varying number of branches.

## C   MORE DISCUSSION ON TRANSFORMER MODEL

### C.1   FORWARD PROCESS

**Proof of Proposition 1**

Without loss of generality, we assume that there are only one head and ignore the head subscript for simplicity. For a specific symbol representation $\boldsymbol{x}$, the query is $q = \boldsymbol{x}^T \tilde{\boldsymbol{Q}}$, the keys are $\boldsymbol{k}_1 = \boldsymbol{x}_1^T \tilde{\boldsymbol{K}}, ..., \boldsymbol{k}_n = \boldsymbol{x}_n^T \tilde{\boldsymbol{K}}$ (i.e., $\boldsymbol{K} = \boldsymbol{X} \tilde{\boldsymbol{K}}$), the values are $\boldsymbol{v}_1 = \boldsymbol{x}_1^T \tilde{\boldsymbol{V}}, ..., \boldsymbol{v}_n = \boldsymbol{x}_n^T \tilde{\boldsymbol{V}}$ (i.e., $\boldsymbol{V} = \boldsymbol{X} \tilde{\boldsymbol{V}}$. We note that $\boldsymbol{q}, \boldsymbol{k}_j, \boldsymbol{v}_j$ are row vectors. Let $(p_1, ..., p_n) = \text{softmax}(\frac{qk_1^T}{\sqrt{d_k}}, \frac{qk_2^T}{\sqrt{d_k}}, ..., \frac{qk_n^T}{\sqrt{d_k}})$.

We note that for symbol $\boldsymbol{x}$, $\boldsymbol{h} = (p_1, \ldots, p_n)\boldsymbol{V}$ and then we have an upper bound on the head representation $\|\boldsymbol{h}\| = \|(p_1, \ldots, p_n)\boldsymbol{X}\tilde{\boldsymbol{V}}\| \leq \max_{i \in [n]} \|\boldsymbol{x}_i \tilde{\boldsymbol{V}}\|$.

Suppose $\|\boldsymbol{x}_i\| = \sqrt{d}$ for all $i \in [n]$, which is reasonable assumption because of the layer normalization. Then $\|\boldsymbol{x}_i \tilde{\boldsymbol{V}}\|^2$ is a Chi-square $\chi^2_{d_v}$ variable. Next we estimate the maximum of $n$ independent Chi-square variables by using the following asymptotic relation. For a random variable $U \sim \chi^2_\nu$ and large $\nu$, we have $\sqrt{2U} - \sqrt{2\nu - 1} \sim \mathcal{N}(0, 1)$. We use the estimation of extreme value of Gaussian variables. For $Z = \max_{i \in [n]} V_i$ and $V_i \sim \mathcal{N}(0, 1)$, we have $\mathbb{E}(Z) \leq \sqrt{2 \log n}$. Based on the above estimation, we have asymptotically $\max_{i \in [n]} \|\boldsymbol{x}_i \tilde{\boldsymbol{V}}\|^2 \lesssim d_v - \frac{1}{2} + \log n + \sqrt{2(2d_v - 1) \log n}$. We can also estimate the maximum of a sequence of $\chi^2$ variables by using the relation that the extreme value of $\chi^2$ distribution (a case of Gamma distribution) asymptotically converges to Gumbel distribution and get a similar result. Hence, we have $\|\boldsymbol{h}\|^2 \lesssim d_v + (\log n - 1 + \sqrt{2(2d_v - 1) \log n})$.

### C.2   BACKWARD PROCESS

We focus on the case with one head and omit the subscript $c$. We assume $d_v = d_k = d_q =: d_h$. We rewrite the forward process as follows. Given the parameters $\tilde{\boldsymbol{Q}} \in \mathbb{R}^{d \times d_h}, \tilde{\boldsymbol{K}} \in \mathbb{R}^{d \times d_h}, \tilde{\boldsymbol{V}} \in$

$\mathbb{R}^{d \times d_h}, \tilde{\boldsymbol{O}} \in \mathbb{R}^{d_h \times d}$.

$$\boldsymbol{X} = (\boldsymbol{x}_1, ...., \boldsymbol{x}_n)^T, \text{ with each row being a symbol embedding, } \in \mathbb{R}^{n \times d} \tag{25}$$

$$\boldsymbol{V} = \boldsymbol{X}\tilde{\boldsymbol{V}}, \in \mathbb{R}^{n \times d_h} \tag{26}$$

$$\boldsymbol{Q} = \boldsymbol{X}\tilde{\boldsymbol{Q}}, \in \mathbb{R}^{n \times d_h} \tag{27}$$

$$\boldsymbol{K} = \boldsymbol{X}\tilde{\boldsymbol{K}}, \in \mathbb{R}^{n \times d_h} \tag{28}$$

$$\hat{\boldsymbol{P}} = \boldsymbol{Q}\boldsymbol{K}^T / \sqrt{d_h} \tag{29}$$

$$\boldsymbol{P} = \text{softmax}(\hat{\boldsymbol{P}}), \text{ with each row being a probability simplex, } \in \mathbb{R}^{n \times n} \tag{30}$$

$$\boldsymbol{H} = \boldsymbol{P}\boldsymbol{V}, \text{ with each row being the head representation, } \in \mathbb{R}^{n \times d_h} \tag{31}$$

$$\boldsymbol{O} = \boldsymbol{H}\tilde{\boldsymbol{O}}, \text{ with each row being the new representation of a symbol, } \in \mathbb{R}^{n \times d} \tag{32}$$

For the backward process, it can be written as follows. Suppose the error signals on the output $\boldsymbol{O}$ is $\boldsymbol{E} = (\boldsymbol{e}_1, ..., \boldsymbol{e}_n)^T$, with each row being an error signal on a symbol representation.

$$\partial \boldsymbol{O} = \boldsymbol{E}, \in \mathbb{R}^{n \times d} \tag{33}$$

$$\partial \boldsymbol{H} = \partial \boldsymbol{O} \cdot \tilde{\boldsymbol{O}}^T, \quad \partial \tilde{\boldsymbol{O}} = \boldsymbol{H}^T \cdot \partial \boldsymbol{O}, \tag{34}$$

$$\partial \boldsymbol{P} = \partial \boldsymbol{H} \cdot \boldsymbol{V}^T, \quad \partial \boldsymbol{V} = \boldsymbol{P}^T \cdot \partial \boldsymbol{H}, \tag{35}$$

$$\partial_V \boldsymbol{X} = \partial \boldsymbol{V} \cdot \tilde{\boldsymbol{V}}^T, \quad \partial \tilde{\boldsymbol{V}} = \boldsymbol{X}^T \cdot \partial \boldsymbol{V}, \tag{36}$$

$$\partial \hat{\boldsymbol{P}} = \partial \boldsymbol{P} \boldsymbol{J}^T, \tag{37}$$

$$\partial \boldsymbol{Q} = \partial \hat{\boldsymbol{P}} \cdot \boldsymbol{K} / \sqrt{d_h}, \quad \partial \boldsymbol{K} = \partial \hat{\boldsymbol{P}}^T \boldsymbol{Q} / \sqrt{d_h}, \tag{38}$$

$$\partial_Q \boldsymbol{X} = \partial \boldsymbol{Q} \cdot \tilde{\boldsymbol{Q}}^T, \quad \partial \tilde{\boldsymbol{Q}} = \boldsymbol{X}^T \cdot \partial \boldsymbol{Q}, \tag{39}$$

$$\partial_K \boldsymbol{X} = \partial \boldsymbol{K} \cdot \tilde{\boldsymbol{K}}^T, \quad \partial \tilde{\boldsymbol{K}} = \boldsymbol{X}^T \cdot \partial \boldsymbol{K}, \tag{40}$$

$$\partial \boldsymbol{X} = \partial_V \boldsymbol{X} + \partial_Q \boldsymbol{X} + \partial_K \boldsymbol{X} \tag{41}$$

We can further compute the expected norm of the gradient as in the proof of Proposition 1 under the gradient independence assumption (Yang, 2019).

### C.3 More about Transformer

Apart from the multi-head attention block, the fully-connected (FC) block is also a multi-branch architecture with sum aggregation. We use $d_{fc}$ to denote the intermediate dimension of fully connected block. The fully connected block consists of two FC layers with weights $W_1^{fc} \in \mathbb{R}^{d \times d_{fc}}$ and $W_2^{fc} \in \mathbb{R}^{d_{fc} \times d}$. In practice, $d_{fc}$ is several times larger than $d$. For example, the "big" Transformer in Vaswani et al. (2017) uses $d_{fc} = 4d$. These sum aggregation operations restrict practitioners to explore Transformer models with large $h$ and $d_{fc}$.

## D More details about experiments

The training procedure of ResNeXt model in Section 5 is the same as Xie et al. (2017). Specifically, we train the models on 8 GPUs with batch size 128, total number of epochs 300 and weight decay 0.0005. We use Nesterov momentum with coefficient 0.9. The learning rate is multiplied by 0.1 at the 150-th and 225-th epoch. For data augmentation, we take a random crop with size 32x32 from a zero-padded 40x40 image or its horizontal flipping.

Now we introduce more details of Transformers on translation tasks in Section 5. The IWSLT14 De-En dataset is collected from Fairseq official site[1], which contains 156K/7K/7K sentence pairs for training/validation/test, respectively. For WMT English-German task, we use **the same data setup as Ott et al. (2018)**. Specifically, we train the model using the training data of WMT16 , which contains 4.5M sentence pairs. **The validation and test sets are newstest13 and newstest14**, respectively.

---

[1]https://github.com/pytorch/fairseq/blob/v0.9.0/examples/translation/README.md

We use head-level dropout which drop heads representation (before the output projection of attention block). The head-level dropout probability is $p$. We set $p = 0.3$ for IWSLT14 De-En task and $p = 0.1$ for WMT16 En-De task, respectively. Our source code is available at anonymous GitHub page[2].

---

[2]https://github.com/AnonymousAKES/STAM-aggregation

