# OpenReview forum: "On the Stability of Multi-branch Network"
_ICLR.cc/2021/Conference — Reject_

### Official Review · AnonReviewer3 · 2020-10-26
**An interesting paper analyzing multi-branch neural networks.**

**Rating:** 4
**Confidence:** 4

**Review:**

This paper studies the multi-branch neural networks. It analyzes the feedforward/backward dynamics of multi-branch networks, proposes an aggregation design named STAM, and shows the feedforward/backward dynamics of multi-head attention layers, and discusses some of its properties.

Here are some issues that might need the authors’ attention.

1. In the introduction, the paper states that simply adding branches multiple times does not often work as expected. I am aware that there are experimental results in the later sections that support this argument. But putting it in the introduction without any concrete examples might be confusing.

2. I think it would also be helpful if there are some brief descriptions of what the proposed STAM does and how it improves over other designs in the introduction.

3. The paper seems to lack a strong main theme and all the contributions and results are loosely connected.

4. The theoretical results are mostly based on the assumption that the weights are randomly initialized. However, the distributions and the properties of the weights might be different after the training begins. Does this imply that the significance of some of the results is limited?

5. The stability analysis is mostly done on norm analysis. The proposed STAM is also inspired by the norm analysis where the goal is to keep the norm of activations and gradients within a reasonable range. However, it seems to me that the theoretical results are a little bit weak: it’s not hard to imagine that the relationship between norms of activations/gradients and the norm of $\alpha$ is somewhat linear, and sum/mean becomes improper when the number of branches grows.

6. Missing words in paragraph 5 of section 3.2: … as the number of branches (see Section 5.2).

7. The aggregation form that the paper has studied is the weighted sum of multiple branches. The paper claims that concatenation is equivalent to sum aggregation for ResNeXt. I’m not fully convinced that this is true considering it’s impractical for a sum aggregation to perform matrix multiplication, batch normalization, and optional ReLU. And I don’t see an easy way to absorb the last 1x1 conv+bn into the branches while satisfying the assumptions of the theoretical results, plus bn will also cancel the effect of $\alpha$ if my understanding is correct.

8. ResNeXt uses the batch normalization layers, which automatically adjust $\alpha$. How is this related to the proposed STAM? Do they serve similar purposes?

Summary:
The paper presents some interesting findings. However, it seems lacking a strong main theme and the novelty/significance of the theoretical results is not strong, either. The proposed STAM only provides incremental design changes to the networks.

Updates:
Thanks for the authors' response and the revisions. However, the paper still lacks strong and significant results. Therefore, I keep my previous rating.

---

> ### Author Response · Authors · 2020-11-24
> **author response**
>
> Thanks for reviewing the paper. Here are responses to your concerns.
>
> 1.	Introduction “adding branches multiple times does not often work as expected.” Confusing.
>
>   Re: Thank you for pointing this out. We modify the presentation and refer to later experiments clearly.
>
> 2.	Present the design of STAM in introduction.
>
>   Re: Thank you for the suggestion. We add more description about STAM in introduction. However the detailed STAM design requires the formal introduction of multi-branch network, which we do not want to expose too much in introduction.
>
> 3.	Lack a strong main theme. Contributions are loosely connected.
>
>  Re: Thank you for pointing this out. The focus of this paper is the training stability of the multi-branch network. However, it indeed involves many pieces, i.e., the theoretical model, practical wisdom, and different architecures. We try to make this more coherent.
>
> 4.	Theory relies on random initialization. Does this imply the results are limited.
>
>  Re: The theory gives strong guarantee for initialization and in the neighborhood of initialization. It is in general still very obscure for the theory of characterizing the training procedure of deep neural network..
>
> 5. The norm analysis seems a bit weak.
>
>  Re: Thanks for pointing this out. It can be derived further about the convergence of SGD based on the forward/backward stability using the analysis tools of NTK.
>
> 6. Missing words in paragraph of section 3.2
>
>  Re: Thanks for pointing out. We corrected this typo.
>
>  7. & 8. 	BN vs $\alpha$ in ResNeXt.
>
>  The batch normalization has the effect to stabilize forward/backward propagation if putting them in appropriate positions, which we have discussed the effect of batch normalization. We do not put $\alpha$ right before the BN, which will cancel the effect of $\alpha$, the same as the reviewer's understanding. In practice, we can use $\alpha$ without normalization layer, or with the post-norm $z_{l+1} = N(z_l + \alpha \cdot\text{multi-branch}(z_l))$ or with the pre-norm $z_{l+1} = z_l + \alpha \cdot \text{multi-branch}(N(z_l))$, where $N(\cdot)$ is the normalization layer.

---

### Official Review · AnonReviewer1 · 2020-10-27
**Borderline paper**

**Rating:** 5
**Confidence:** 3

**Review:**

This study focuses on the stability of multi-branch networks. It analyzes the forward and backward stability of multi-branch network, and builds the relations with some widely-adopted initialization and normalization schemes. A simple new aggregation method is proposed that enjoys better stability than the sum and average aggregations. The method is extended to the multi-head attention layer in Transformer. Experiments on image classification and machine translation tasks using ResNeXt and Transformer are conduced to show the efficacy.

Pros:
The paper is well-written. Its motivation is clear and the analyses are technically correct. The proposed method is well interpretable and easy to follow. The connections with some initialization and normalization methods are well explained.

Cons & questions:
1.	The paper should offer the definition of “stability” first. Does it refer to the numerical stability of features or the robustness to noise? What does $\epsilon$ in Eq (3) mean? Is it the perturbation to the input feature $h_{in}$?
2.	How to quantize stability? The experiments on image classification and machine translation show that the proposed method STAM is able to improve classification accuracy and BLEU score. I think these metrics are more related to representation ability. How do we know the stability is improved?
3.	The extendibility is limited. The analyses are based on the assumption that all branches have the same structure without normalization layers. Does it work for network with variant branches? How to generalize this study to network with BN layers? What about the results of experiment 5.1 when BN layers are enabled? Since BN layers have been successful in solving the training difficulties of deep networks (not only multi-branch), could the authors explain the superiority of this study over BN in practical implementation? Besides, are the analyses and proposed methods also applicable to concatenation aggregation?
4.	The improvements on CIFAR-10 and CIFAR-100 over baseline are insignificant.

---

> ### Author Response · Authors · 2020-11-24
> **author response**
>
> Thanks for reviewing the paper. Here are responses to your questions.
>
> 1.	Definition of stability. And quantization
>
>  Re: The stability is described that the forward/backward signal can be efficiently propagated. This is not related with numerical stability of features or robustness to noise. The $\epsilon$ in Eq (3) is to deal with the randomness of initialization.  This stability has been used for many papers in characterizing the trainability of deep neural network.
>
> 2.	Normalization layers, concatenation.
>
>  We have discussed the effect of normalization layers, scaling down the initialization and concatenation in Section 3.2. It is for simplicity of theoretical presentation to assume same structure in all branches, which is already enough to show the key property of multi-branch network.
>
> 3.	CIFAR improvement is insignificant.
>  Our model has the same parameter as ResNeXt. We want to demonstrate the effect of aggregation approach. The improvement on CIFAR100 is almost 0.9 point over very strong baseline ResNeXt, which should be viewed significant for such small modification.

---

### Official Review · AnonReviewer2 · 2020-10-29

**Rating:** 3
**Confidence:** 4

**Review:**

[Summary]

This paper studies the training of multi-branch networks, i.e. networks formed by linearly combining multiple disjoint branches of the same architecture. The core contribution in this paper is the “STAM” aggregation rule which is to set the combination coefficient to $1/\sqrt{C}$ for a network with $C$ branches. This aggregation rule is justified by (1) theoretical analysis on the function values and gradient norms at initialization, and (2) experiments on residual networks and transformers showing that this rule performs better than the baseline rules (such as sum or average).

[Pros]

--- The problem this paper focuses on (multi-branch architectures) is important and I think worth more studies since they are used more widely these days. From a theoretical perspective, the number of branches seems like a similar thing as the width of a network, yet the understanding is not yet as comprehensive as our understanding on the width (for which we know things like what are their effects, how should we initialize a wide network, etc).

--- The experimental results are interesting and could be a good reference point for designing these multi-branch architectures. Specifically, it is interesting to see it is better to explicitly do a fixed $1/\sqrt{C}$ scaling, rather than making it trainable or embedding it into the individual linear layers in each block.

--- The paper is generally clearly written and easy to follow. The notation between the theory and empirical parts matched quite well.

[Cons]

--- I feel like the core contribution of the $1/\sqrt{C}$ scaling rule is rather straightforward and not necessarily justified as a new method or theory, especially with the analogy to width in mind: It is standard to initialize weight matrices to have scale $1/\sqrt{d}$ where $d$ is the either the input or output dimension as this gives the right normalization. Even with the other “baselines” (making the $1/\sqrt{C}$ trainable or embedding it into the individual blocks) performing worse according to this paper, I still feel like a researcher or practitioner who encounters the issue of choosing the aggregation rule could still end up with the same solution proposed in this paper.

--- The theoretical results also build on the exact same normalization idea ($1/\sqrt{C}$ scaling gives the right normalization) as for width, and do not seem to provide us with much new theoretical understandings for what is unique about the multi-branch architecture. From my (maybe biased) perspective, I would have been more interested in some more in-depth understandings (either theoretical or empirical), such as what is the effect of adding more branches (and where does diminishing return kick in). Reading this paper leaves me more questions than answers about these multi-branch networks.

------
After rebuttal: Thank the authors for the response. I think my original concern largely stands and would like to keep my original evaluation of the paper.

---

> ### Author Response · Authors · 2020-11-24
> **author response**
>
> Thanks for reviewing the paper. Here are responses to your concerns.
>
> 1. The contribution is straightforward and not necessary. The initialization has already done the job.
>
>  Re: We have discussed the effect of initialization in Section 3.2. We did the experiments and reported the result. Although the result may not be able to change researcher or practitioners' choice, our paper presents a more in-depth understanding on the multi-branch network, which should be appropriately valued.
>
> 2.	Theory does not provide new understanding of multi-branch architecture. What is the diminishing return for adding more branches?
>
>  Re: It is definitely worthy to think more on the reviewer's suggestion. However, in this paper, we are unveiling different aspects of the network design. We focus the training stability as the number of branches goes larger based on theoretical characterization while EfficientNet discloses a performance scaling law of depth, width and resolution based on large sets of experiments. It is in general hard to formulate the diminishing return in a theoretical framework.

---

### Official Review · AnonReviewer4 · 2020-10-29
**Good intuition, not convincing experiments**

**Rating:** 5
**Confidence:** 4

**Review:**

##### Summary
This paper proposes to scale down the features of multiple branch networks by $1/\sqrt{C}$ during aggregation where $C$ is the number of branches. The authors provide some theoretical insights to back up the method. However, the experiments are not very convincing. I had a good impression of this paper when I read the method sections but until the experiment sections. There are many problems in the experimental setup that makes me hesitate to accept this paper. The authors have a very unclear definition of the number of branches which gives me the impression that the authors just introduce a hyperparameter $\alpha = 1/\sqrt{C}$ and they can pick whatever $\alpha$ or $C$ they want in the experiments tailored for the story.

##### Strength
- The arguments and theory in sections 3 & 4 are sound and interesting.
- The proposed method is quite simple.
- The proposed method leads to improvements on CIFAR 10 & 100.
- By tuning the model architecture with the proposed method, the authors also show improvements on IWSLT 14 & nestest2014.


##### Weaknesses

- This paper is lack of comparisons with a series of related works that sets the outputs of residual branches to zeros such as FixUp (Zhang et al., 2019) and Re-Zero (Bachlechner et al., 2020).
- It is very suspicious that the authors do not report the BLEU (+STAM) of the first row in Tables 2 & 3. It gives me the impression that the authors try to hide the fact that the proposed method does not do well in this setting and they have to tune the model structure and training settings to find some cases that the proposed method works.
- In Table 3, the baseline (Ott et al., 2018) is trained for about 43 epochs. In my experience, it starts to overfit after that. However, the authors train all their models for 150 epochs. Otherwise, it is possible that the Transformers without STAM BLEU converge faster and overfit at epoch 150, which makes them look worse than the ones with STAM. Based on Appendix D, the authors also use a new technique head-level dropout which was not used by Ott et al., 2018. I would like to see the authors use the same setting as the baseline or report curve of BLEU scores at different epochs to convince me that this is not the case.
- The author set $\alpha = 1/\sqrt{C/8}$ for Transformers and $\alpha = 1/\sqrt{C/4}$ for ResNeXt and just provide some very hand waiving explanation, which gives an impression that the theory doesn't work and all we need is to tune a hyperparameter $\alpha$.
- On page 6, "We give an upper bound on the forward process when the softmax behaves more like a “max” operator, which is usually the case after initial training." This is a false claim. It is more often that the attentions assign large weights to multiple positions. Making the attention sparse usually restricts the models and leads to worse results.
- The claim in Appendix C.3 makes the authors' definition of $C$ very arbitrary. The author claim that the 2-layer FFN with $d_{fc} = 4d$ in a Transformer-big has 4 branches. I don't see why there is a reason that this should view as having four branches. This should be considered as having only one branch only. Otherwise, since it has only 2 layers, we can view each hidden neuron in the middle layer as one branch. The authors just take their advantage to group ever $d$ neurons as one branch, but some other people can also group them in any other ways and say there are $C$ branches take doing them arbitrarily. For example, I can group ever 4 neurons as one branch and say this FFN has $d$ branches, or even group the first $2d$ dimensions as one branch and the rest of the neurons as $2d$ branches and say this FFN has $2d + 1$ branch. Overall, this makes me feel that the authors are playing a game of definitions and choosing whatever $C$ or $\alpha$ as they want.


##### Questions & Suggestions
- How does STAM work when applied to depthwise separable convolution in MobileNets and EfficientNets which can be viewed as an extreme case of multi-branch convolution.
- It would be more exciting to see if this method can be applied to the field of neural architecture search (NAS) where people explicitly train a gigantic multi-branch network and has problems with having some branches undertrained.

##### References
- Zhang, Hongyi, Yann N. Dauphin, and Tengyu Ma. "Fixup initialization: Residual learning without normalization." arXiv preprint arXiv:1901.09321 (2019).
- Bachlechner, Thomas, et al. "Rezero is all you need: Fast convergence at large depth." arXiv preprint arXiv:2003.04887 (2020).

---

> ### Author Response · Authors · 2020-11-24
> **author responses**
>
> We thank the reviewer for the detailed comments. Here are 1-1 responses to the reviewer's questions.
>
> 1. Lack of comparison with FixUp and Re-Zero.
>
>  Re: Fixup and ReZero are for residual network with large depths, while we study the multi-branch structure. Although the scaling factor are at the same position if using residual structure, we are targeting different problems.
>
>  1) Fixup scales down the initialization to train deep ResNet without BN. We have discussed the initialization approach for stabilizing multi-branch network in Section 3.2. We argued that scaling down initialization can have similar effect of stabilization as  STAM  theoretically and proposed a new initialization scaling factor based on backward stability. 2) ReZero introduces a learnable $\alpha$ on the  output of residual block. The experiments in ReZero focus on the fast training but do demonstrate test/valid improvement over baselines.
>
> 2. Report BLEU on baseline setting + STAM in Table 2&3. It gives me the impression that the authors try to hide the fact that the proposed method does not do well in this setting and they have to tune the model structure and training settings to find some cases that the proposed method works.
>
>  Re: We are not hiding anything. For baseline settings, they are stable itself by initialization and the hyperparameter setting.  We do not use extra $\alpha$, or equivalently $\alpha=1$.
>
>  In the experiments,  we use fixed $\alpha$ and do not tune this value because we are not trying to find the best model structure and tune the hyperparameters. Instead, we want to show the effect of adding more branches and how the STAM aggregation can mitigate the issue of instability and fully train the network to its potential as the number of branches becomes large.
>
> 3. In Table 3, the baseline (Ott et al., 2018) is trained for about 43 epochs. In my experience, it starts to overfit after that....
>
>  Re: Thanks for the insightful comments. As using more branches, we indeed observed overfitting and introduced the head-level dropout to effectively reduce the validation loss. This is something trick to battle over the overfitting, and is not that relevant to our theme "stability of multi-branch network". We will make this clear in the revision.
>
> 4. The author set $\alpha$ for Transformers and  for ResNeXt and just provide some very hand waiving explanation, which gives an impression that the theory doesn't work and all we need is to tune a hyperparameter .
>
>  Re: The theory is based on that each branch can stably forward/backward propagate itself. The experiment settings are consistent with the theory. For the ResNeXt, the  number of branches is the same as original ResNeXt. We use $\alpha=1/\sqrt{C}$ to aggregate in our ResNeXt.
>
>  For Transformers, the original structure is stable because of initialization and the hyperparameter setting. We use a corresponding mini-version as one branch. We set the $\alpha$ accordingly. In the original submission, we use $C$ to denote the number of heads, which arouses confusions. In the revision, we clarify this by using $h$ to denote heads and use $C$ to denote the effective number of branches. Moreover, in the experiments, we want to demonstrate that the scaling factor $\alpha$ should  be inversely proportional with the root of the number of branches, which aligns with the theoretical claim, rather than some exact value $1/\sqrt{C}$. We set $\alpha=1/\sqrt{C}$ for simplicity.
>
> 5. On page 6, "We give an upper bound on the forward process when the softmax behaves more like a “max” operator, which is usually the case after initial training." This is a false claim.
>
>  Re: Thank you for pointing out this. The statement is a relative description compared to the initialization. However, we agree with reviewer's concern and modify the draft.
>
> 6. The claim in Appendix C.3 makes the authors' definition of  very arbitrary...
>
>  Re: We thank the reviewer for pointing out this. We admit that the definition of branch need to be clarified. From the theory, a branch itself should be a stable forward/backward stability for a constant lr optimizer. For a 2 layer FFN with $d_{fc}=4d$ and a fixed $d$, we can view it as either one branch or four branches. The view needs to change when the intermediate dimension goes extremely wider, e.g. from 4 to 1000, 10000, 100000. The reviewer may not agree with this because such setting is not used in practice. However, we focus on theoretical consistency of multi-branch network as $C$ varies, which is important for us to define the branch. Note that a constant learning rate SGD will fail to train a 2 layer FFN eventually when the intermediate layer goes to infinity.
>
> 7. Apply to EfficientNet and NAS.
> Thanks for the suggestion. It seems that there are works using a tunable scalar in NAS and our theory can serve as a basis for this practice.

---

### Decision · Program_Chairs · 2021-01-07
**Final Decision**

**Decision:**

Reject

**Comment:**

This paper studies the training of multi-branch networks, i.e., networks formed by linearly combining multiple disjoint branches of the same architecture.  The four reviewers seem to reach a consensus that the paper is not ready for publication for ICLR.